


# Contribution of Rock Glacier Discharge to Late-Summer and Fall Streamflow in the Uinta Mountains, Utah, USA

Jeffrey S. Munroe[1], Alexander L. Handwerger[2,3]

[1] Geology Department, Middlebury College, Middlebury, 05753, USA

[2] Joint Institute for Regional Earth System Science and Engineering, University of California, Los Angeles, 90095, USA

[3] Jet Propulsion Laboratory, California Institute of Technology, Pasadena, 91109, USA

*Correspondence to*: Jeffrey S. Munroe (jmunroe@middlebury.edu)

**Abstract.** Water draining from rock glaciers in the Uinta Mountains of Utah (USA) was analyzed and compared with samples of ground water and water from the master stream in a representative 5000-ha drainage. Rock glacier water resembles snowmelt in the early summer, but transitions to higher values of *d-excess* and greatly elevated Ca and Mg content as the melt season progresses. This pattern is consistent with models describing a transition from snowmelt, to melting of seasonal ice, to melting of perennial ice in the rock glacier interior in late summer and fall. Water derived from this internal ice appears to have been the source of ~25% of the streamflow in this study area during September of 2021. This result emphasizes the significant role that rock glaciers can play in the hydrology of high-elevation watersheds, particularly in summers following a winter with below average snowpack.

**Keywords**: Rock glacier; hydrology; permafrost; stable isotopes; climate change

## 1 Introduction

Contemporary climate change is responsible for an array of dramatic effects in high mountain environments (Adler et al., 2019; Chakraborty, 2021). Average temperatures of air (Bonfils et al., 2008; Minder et al., 2018) and permafrost (Biskaborn et al., 2019) are rising, glaciers are retreating (Sakai and Fujita, 2017; Sommer et al., 2020), the ranges of plants (Alexander et al., 2018; Albrich et al., 2020) and animals (Millar and Westfall, 2010; Rödder et al., 2021) are shifting, and ecosystem services (Egan and Price, 2017; Palomo, 2017) and the societies that depend on them (McDowell et al., 2019; Xenarios et al., 2019) are in a phase of readjustment. Documenting and understanding these changes is of crucial importance in mitigating natural hazards (Stoffel and Corona, 2018; Thaler et al., 2018), anticipating future scarcity of water resources (Beniston et al., 2018; Rowan et al., 2018), designing appropriate conservation strategies (Catalan et al., 2017), and planning for a future in which mountain environments look and function differently than they have for the past century (Huss et al., 2017).



A component of mountain landscapes with strong potential to document past and present environmental

changes, and a notable vulnerability to climatic perturbations, are features known as rock glaciers. Typically present

in cold environments that are too dry for the formation of ice glaciers, rock glaciers are mixtures of rock debris and

perennial ice that move downslope through a combination of creep and basal shear (Wahrhaftig and Cox, 1959;

Giardino et al., 1987; Giardino and Vitek, 1988). Given their genesis, their composition, and their behavior, rock

glaciers exist at the intersection of climate, the cryosphere, and hydrology.

Traditionally, rock glacier research focused on the distribution and paleoclimatic significance of these

features (Konrad et al., 1999; Johnson et al., 2021). Modern updates to these investigations are applying high

precision GPS (Buchli et al., 2018), photogrammetry (Kenner et al., 2018), surface-exposure dating (Lehmann et al.,

2022), and remote sensing to monitor rock glacier movement (Strozzi et al., 2020), offering an unprecedented

understanding of the relationship between rock glacier behavior and climate change. Studies have also sought to

explore the role of rock glaciers as refugia for cold-adapted organisms in the face of warming temperatures (Millar

et al., 2015; Brighenti et al., 2021).

An additional line of inquiry with critical importance in regions characterized by water scarcity is the

contribution of rock glaciers to high mountain hydrology (Rangecroft et al., 2015; Jones et al., 2019). The

interconnected pore space within the typically coarse debris comprising a rock glacier allows these features to serve

as aquifers, storing and releasing water over a variety of timescales (Geiger et al., 2014; Harrington et al., 2018;

Wagner et al., 2020; Halla et al., 2021). Moreover, perennial ice within the interior of an active rock glacier is a

reservoir of longer-term storage that is nonetheless vulnerable to being lost from the system through melting in

excess of new ice formation. Studies have investigated the ice content of rock glaciers using geophysical methods

like ground penetrating radar and invasive approaches like drilling (Krainer and Mostler, 2002; Krainer et al., 2015;

Petersen et al., 2020; Wagner et al., 2021). Extrapolation from these investigations, and incorporation of empirical

transfer functions, has supported estimates of rock glacier water storage for some areas (Azócar and Brenning, 2010;

Rangecroft et al., 2015; Janke et al., 2017; Jones et al., 2018). Nonetheless, uncertainty remains about how much ice

is stored within rock glaciers, the vulnerability of this ice to climate warming, and how much ice may already be

melting and contributing to base flow, particularly in late summer after the melting of seasonal snow has ceased.

Here we investigate the water draining from representative rock glaciers in the Uinta Mountains in

northeastern Utah, a mountain range in which rock glaciers have been inventoried (Munroe, 2018) and monitored

(Brencher et al., 2021) in previous work. Automated samplers were used to collect time series of water discharging

from two rock glaciers, a non-rock glacier spring, and along the master stream. All samples were analyzed for

cation chemistry and stable isotopes to test two related hypotheses: 1) that the rock glacier springs would exhibit

properties distinct from the other water sources and consistent with the melting of internal ice in late summer; and 2)

that late summer streamflow along the master stream would contain a non-trivial amount of rock glacier-sourced

water.



## 2 Study Area

The study area for this project is in the upper West Fork Whiterocks River watershed in the southeastern sector of the Uinta Mountains (Figure 1). The watershed has an area of ~5000 ha above the lowest sampling site, and elevations range from 2960 to over 3700 m. The Uinta Mountains (hereafter, the "Uintas") are a substantial component of the Rocky Mountain system located in northeastern Utah in the western United States. The Uintas are the highest mountains in this region, reaching elevations in excess of 4 km. Geologically, the bedrock of the Uintas is a thick sequence of Precambrian siliciclastic rocks that was uplifted during the Laramide orogeny beginning in the early Paleogene (Sears et al., 1982; Hansen, 1986; Dehler et al., 2007). Pleistocene valley glaciers eroded deep cirques and glacial canyons, and deposited massive lateral and end moraine systems (Atwood, 1909; Munroe and Laabs, 2009). No glaciers remain in the Uintas today, however the climate at higher elevations, where mean annual temperatures are <0 °C (Munroe, 2006), supports patterned ground, talus, and abundant rock glaciers. Previous work using optical imagery (Munroe, 2018) and satellite-based radar interferometry (Brencher et al., 2021) identified more than 200 active rock glaciers in the Uintas, and many more rock glaciers that are no longer moving.

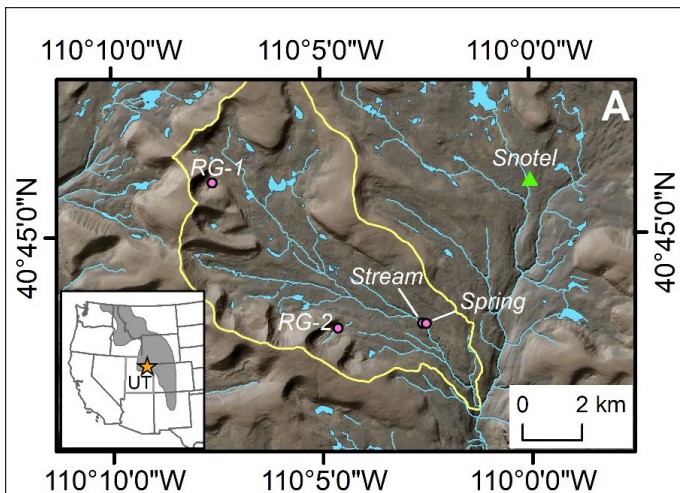

**Figure 1**: Location Map of the study area. Inset shows location of the Uinta Mountains (orange star) within the state of Utah (UT). Gray shaded polygon represents the Rocky Mountains. Map presents the upper Whiterocks River watershed (yellow boundary), the locations of the RG-1, RG-2, Stream, and Spring water samplers (pink circles), and the Chepeta SNOTEL site (green triangle).

## 3 Methods

This project centered on the collection of time series of water samples using automated samplers outfitted with a carousel of 24 bottles. The samples were deployed with a solar powered battery system, allowing them to run



throughout the summer with no maintenance.  To reduce the possibility of isotopic fractionation related to
evaporation, sample bottles were modified following published methodology (von Freyberg et al., 2020).   In each
location, the samplers were deployed in a position higher than their water intake to facilitate free draining of the
intake hose between samples.  The weighted strainer on the end of the water intake line was wrapped in 100-μm
nylon mesh to prevent coarse material from clogging the sampler pump.  Each sampler was programmed to collect a
45-mL sample twice each day, at midnight and noon.  For three days these samples (six samples total) were
composited in a single bottle, thus the 24 bottles in each sampler represented a maximum deployment duration of 72
days.
Two samplers were deployed at springs discharging from the base of rock glaciers that were the focus of
previous investigations (Munroe, 2018).   These features, "RG-1" and "RG-2", are typical of cirque floor, tongue-
shaped rock glaciers in the Uinta Mountains (Figures 1 and 2).

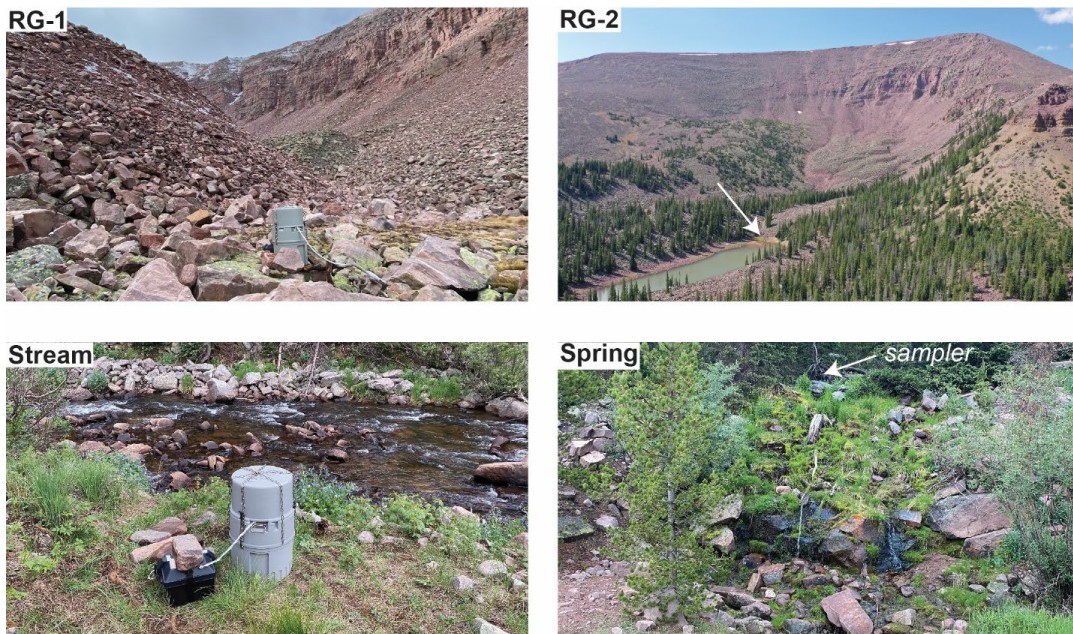


**Figure 2**: Pictures of water samplers at RG-1,  RG-2, the Stream, and the Spring sites.

Each is approximately 600 m long, 100 m wide, and has steep frontal and side slopes standing up to 20 m tall.  Fresh
exposures on these slopes reveal that the rock glaciers consist of several meters of coarse, openwork boulders
overlying a diamicton with a sand matrix.   Internal ice is not exposed in either rock glacier, however data loggers
reveal that the springs maintain a temperature of 0 °C throughout the summer, and rock glacier surface temperatures
equilibrate at -5 °C or colder beneath winter snow cover (Munroe, 2018).  Satellite InSAR (interferometric synthetic
aperture radar) analysis indicates that these features move slowly during the winter and accelerate during the





summer to velocities of ~10 cm/yr (Brencher et al., 2021).  Collectively these observations suggest the presence of
ice within the rock glacier interior.  The spring sampled at RG-1 has a typical summer discharge of 15 L/min.  The
discharge at RG-2 was not measured directly, but a water-level logger records diurnal fluctuations of 0.2 (early
summer) to 0.02 m (late summer) of the lake into which the spring flows.  Given the surface area of the lake (12,000
m$^2$), these daily variations suggest a discharge on the order of $10^2$ to $10^3$ L/min.

Two other water samplers were deployed at non-rock glacier locations.  The "Spring" sampler collected

groundwater discharging from a typical spring unrelated to a rock glacier, and the "Stream" water sampler was
positioned along the main channel of the West Fork Whiterocks River, the master stream in this drainage (Figure 1).
These samplers were configured in an identical manner to those deployed at the rock glacier springs.

To constrain the properties of precipitation in the study area, grab samples of snow were collected on the

surfaces of RG-1 and RG-2 when the water samplers were deployed.  Water draining from a melting snowbank on
RG-2 was also collected.  Rain was collected during the deployment period at the RG-2 and the Spring locations
using samplers of a design intended to eliminate evaporation-related fractionation of isotope values (Gröning et al.,

2012).

All samplers were installed at the beginning of July, 2021 (Table 1).  At the Stream, Spring, and RG-2

samplers a subsample was taken from the first bottle about a week later, with the remainder left inside the sampler.
This procedure provided a check on the potential role of evaporation fractionating the water samples as they waited
inside the sampler.  All bottles were emptied at the beginning of September, and the samplers were relaunched to
run until mid-October, when they were emptied again and deactivated for the winter.  The two precipitation
samplers were emptied when the water samplers were serviced.  All samples for stable isotope analysis were filtered
in the field to 0.2 μm and stored in 7-ml glass vials with Teflon-lined caps.  Samples for ICP-MS analysis were
stored in 15-ml centrifuge tubes. These samples were vacuum filtered with Whatman Number 1 paper in the lab and
acidified to pH 2 with trace-element grade $HNO_3$.  In a preliminary phase of this project, daily samples were also
collected at the RG-2 spring in the fall of 2020.

| Table 1. Locations, Dates, and Durations of Water Sampler Deployments | | | | | | | |
|---|---|---|---|---|---|---|---|
| Sampler | Latitude | Longitude | Elevation (m) | Deployed | Emptied | Emptied | Duration (Days) |
| RG-1 | 40.766906 | -110.127608 | 3408 | 7/2/2021 | 9/5/2021 | 10/7/2021 | 97 |
| RG-2 | 40.721883 | -110.076875 | 3197 | 7/1/2021 | 9/2/2021 | 10/6/2021 | 97 |
| Spring | 40.723016 | -110.042131 | 2977 | 7/3/2021 | 9/2/2021 | 10/6/2021 | 95 |
| Stream | 40.722979 | -110.043123 | 2965 | 7/3/2021 | 9/6/2021 | 10/6/2021 | 95 |


Stable isotope measurements were made with a Los Gatos 45-EP Triple Liquid Water Isotope Analyzer at

Middlebury College.  Samples were run against a bracketing set of 5 standards and calibrated with a cubic spline
function. Each sample was analyzed 10 times, with the first 2 injections discarded to minimize cross-over.



Standards were run as unknowns after every five samples as an internal check on the results. Results were
compared with the Global Meteoric Water Line-GMWL (Craig, 1961) as well as a Local Meteoric Water Line
(LMWL) estimated from OIPC, the Online Isotopes in Precipitation Calculator (Bowen and Wilkinson, 2002;
Bowen and Revenaugh, 2003). Values of *d-excess* were calculated as *d-excess* = $\delta D-(8*\delta^{18}O)$ (Dansgaard, 1964).
Hydrochemical characterizations were made with a Thermo iCap ICP-MS at Middlebury College. Samples
were run against a set of standards derived from NIST Standard Reference Material 1643f "Trace Elements in
Water". An in-house standard was used to determine the abundance of Si and Ti, which are not present in 1643f.
The NIST standard and the in-house standards were run after every 10 unknowns and a linear correction was applied
to compensate for instrument drift. Interpretation focused on elements that consistently exhibited concentrations >1
ppb.

**4 Results**
A total of 141 water samples was analyzed, consisting of 134 samples from the four time-series (including the 3
duplicates), 4 samples of rain, 2 samples of snow, and 1 sample of snow melt. The time-series are essentially
complete with no gaps between early July and mid-October. The lone interruption is one bottle from the Stream
sampler, representing 18-20 July, that was empty, apparently because the water level in the river briefly dropped
below the position of the intake hose.
Overall values of $\delta D$ in the time-series range from -118.94 to -83.71‰. Values of $\delta^{18}O$ range from -16.36
to -12.24‰, and $\delta^{17}O$ from -9.13 to -6.39‰ (Table 2). The mean of $\delta D$ is lowest in the Spring samples (-113.44‰)
and highest at RG-1 (-91.24‰). The same pattern holds for mean values of $\delta^{18}O$ and $\delta^{17}O$ (Table 2). Values of *d-*
*excess* are highest at the RG sites, and lowest (~10‰) in the Stream (Table 2). Values of $\delta D$ and $\delta^{18}O$ for the
subsamples from the first bottle in the Stream, Spring, and RG-2 samplers are quite similar to the remainder that was
left inside the collector during the summer (Figure 3).

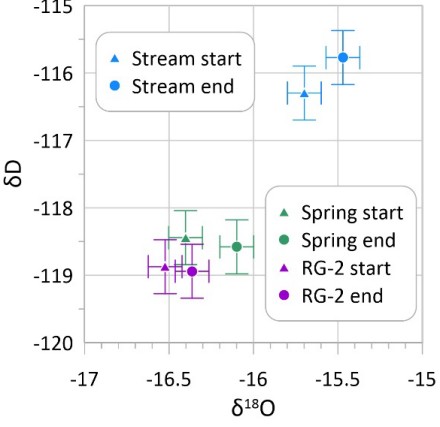

**Figure 3**: Comparison of isotope values measured for samples from RG-2, the Stream, and the Spring samplers. Subsamples were removed from the first sample in early July and the remainder of the water was left inside the collector until early September. Analysis of the sample pairs confirms that potential evaporation-related fractionation was minimal.



| Table 2: Isotope Values for Water Samples | | | | |
|---|---|---|---|---|
| | $\delta^2H$ | $\delta^{18}O$ | $\delta^{17}O$ | d-Excess |
| | (‰) | (‰) | (‰) | (‰) |
| **Stream (n=31)** | | | | |
| Mean | -103.86 | -14.17 | -7.38 | 9.52 |
| Median | -102.55 | -13.90 | -7.40 | 9.02 |
| Standard Deviation | 4.92 | 0.57 | 0.31 | 1.52 |
| Minimum | -115.77 | -15.47 | -8.06 | 7.97 |
| Maximum | -97.05 | -13.52 | -6.78 | 12.62 |
| **Spring (n=33)** | | | | |
| Mean | -113.44 | -15.57 | -8.80 | 11.16 |
| Median | -113.59 | -15.60 | -8.84 | 10.79 |
| Standard Deviation | 3.25 | 0.42 | 0.20 | 1.03 |
| Minimum | -118.58 | -16.12 | -9.13 | 9.57 |
| Maximum | -103.35 | -14.28 | -8.30 | 13.80 |
| **RG-1 (n=33)** | | | | |
| Mean | -91.24 | -13.13 | -6.94 | 13.83 |
| Median | -87.22 | -12.80 | -6.62 | 13.87 |
| Standard Deviation | 8.52 | 0.88 | 0.60 | 2.59 |
| Minimum | -113.35 | -15.67 | -8.48 | 9.80 |
| Maximum | -83.71 | -12.24 | -6.39 | 20.82 |
| **RG-2 (n=34)** | | | | |
| Mean | -101.32 | -14.53 | -7.52 | 14.93 |
| Median | -98.27 | -14.24 | -7.19 | 15.50 |
| Standard Deviation | 7.46 | 0.79 | 0.58 | 1.32 |
| Minimum | -118.94 | -16.36 | -8.76 | 11.97 |
| Maximum | -89.99 | -13.10 | -6.92 | 16.91 |

Values of $\delta D$ and $\delta^{18}O$ are linearly and significantly (p<0.001) related with a slope of 8.8 and a Y-intercept of 24.4‰ (Figure 4). Lower values of $\delta^{18}O$ plot closer to the GMWL; higher values of $\delta^{18}O$ plot increasingly above the GMWL. Plotting the data from the individual samplers separately, with color coding by month, reveals additional details (Figure 5). Values for the Stream and Spring samplers plot along the GMWL through the summer. For the Stream, the lowest values are from July with higher values in late summer and fall. For the Spring, the lowest values are again July, with the highest values in August; September and October values fall in between (Figure 5). On the other hand, for the two rock glaciers, July values are again lower and closer to the GMWL, but values from late summer and the fall plot notably above the GMWL with *d-excess* up to 20‰. At RG-2, a similar pattern was noted in daily samples collected during September, 2020 (Figure 5).



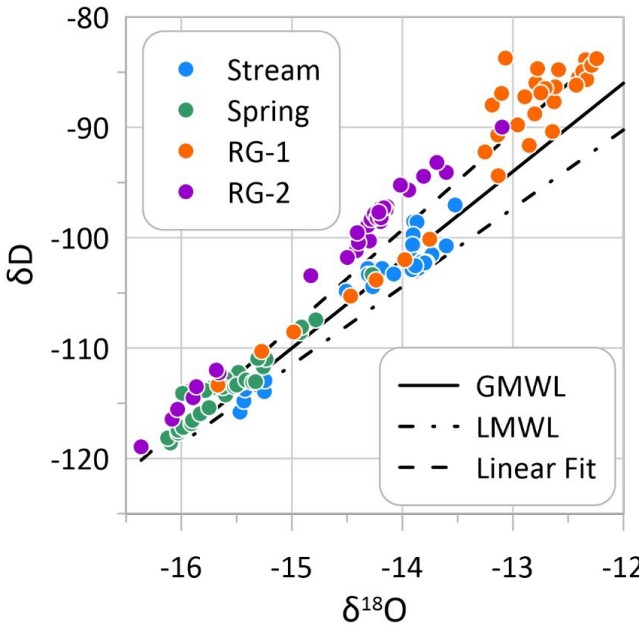

**Figure 4**: Dual isotope plot of $\delta^{18}O$ and $\delta D$ for the samples collected at RG-1, RG-2, the Stream, and the Spring. The Global Meteoric Water Line (GMWL), a local meteoric water line (LMWL) determined from the Online Isotopes in Precipitation Calculator (Bowen and Wilkinson, 2002; Bowen and Revenaugh, 2003), and a linear fit to the data are presented for reference.


Plotting the data from the different samplers as time-series reveals patterns in the evolution of isotope

values during the sampling period (Figure 6). Given the strong correspondence between values of $\delta D$ and $\delta^{18}O$,
only $\delta^{18}O$ is presented for clarity. Values are low at the start of the sampling period (early July), and generally rise
in all records through the summer and early fall (Figure 6A). The Spring and RG-2 both start below -16‰; the
Stream and RG-1 start slightly higher, near -15.5‰. All of the records exhibit transient spikes to less negative
values that occur quickly and taper gradually back to background levels (Figure 6A). These spikes align with pulses
of precipitation recorded at the Chepeta SNOTEL (snowpack telemetry) site <10 km to the north, and at a similar
elevation (Figure 1). Thus, it is likely that they represent rainstorms that delivered water less depleted in $\delta^{18}O$
relative to SMOW, a response reported in other studies (Krainer and Mostler, 2002). After these pulses are removed
from the data to highlight the background trends at each of the sites (Figure 6B), the record from the Spring is seen
to be the most stable, with nearly all values between -16 and -16.5‰. The water at RG-2, which started off similar
to the Spring, rises steadily to a maximum of -14‰ in early October. The Stream rises from -15.5‰ to -14‰ by the
third week of August, and stabilizes through the end of the record. Finally, RG-1, which also starts at -15.5‰, rises
rapidly in the first half of July, then more gradually until early September, when it peaks at -12.5‰ before dropping
to -13‰.





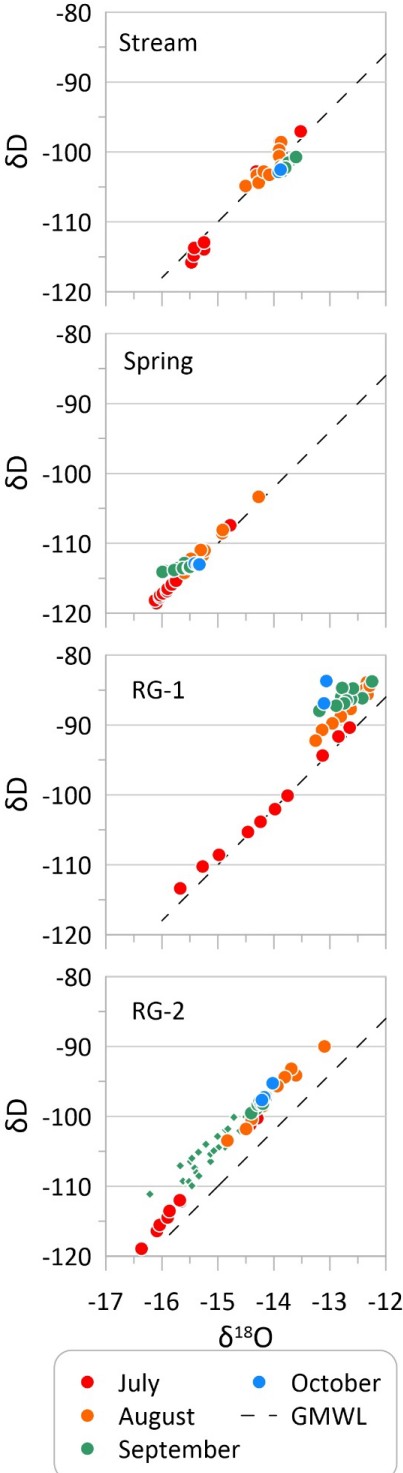

**Figure 5**: Dual isotope plots for the four individual time series. Color-coding represents the month of sample collection. The tendency for samples at the Stream and Spring to remain on the waterline while samples from the rock glaciers deviate to higher values of *d-excess* in late summer and fall is clearly evident. Green diamonds for RG-2 present reconnaissance data from September, 2020.

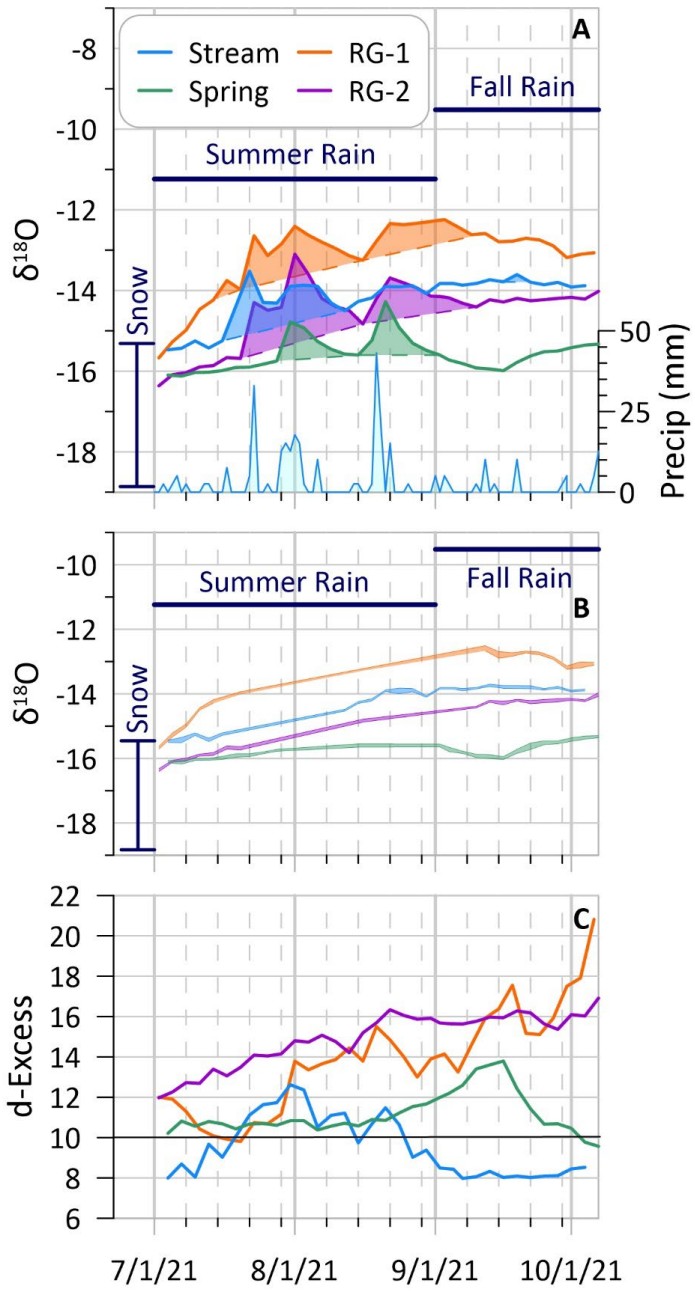

**Figure 6**: Time series from the four sampling sites. (A) Values of $\delta^{18}O$ presented along with average values for snow and rain, and daily precipitation recorded at the Chepeta SNOTEL site (Figure 1). (B) Same as Panel B with transient spikes in $\delta^{18}O$ due to precipitation events removed. (C) Times series of *d-excess*. The reference value of 10‰ is highlighted.



Values of *d-excess* in the time-series exhibit varying patterns (Figure 6C). Values from the Stream initially rise,
then fall through August and stabilize at 8‰ in the fall. The Spring samples are initially stable between 10 and
11‰, then rise in early September to a high of 14‰, before falling back to 10‰. The two rock glaciers sites, in
contrast, rise steadily from near 10‰, to ≥17‰ in early October (Figure 6C).

Context for the isotope values from the water samplers is provided by the precipitation samples collected at
the Spring and RG-2 sites, and grab samples of snow from RG-1 and RG-2 (Figure 6). Values of $\delta^{18}O$ in snow are
low, averaging -17‰. In contrast, bulk precipitation falling in July and August at the Spring and RG-2 averages -
11.2‰, and fall precipitation averages -9.5‰. Values of *d-excess* in snow samples are ~10‰, whereas values in
rain samples are 13 to 21‰.

Hydrochemical analysis with ICP-MS reveals 12 elements that are consistently detectable in these samples:
Ba, Ca, Fe, K, Mg, Mn, Na, Ni, Rb, Si, Sr, and Ti. Ca and Si are generally the most abundant cations, with mean
abundance ~1500 to 2000 ppb, followed by K, Na and K with abundances averaging 500-800 ppb. Fe and Ba are
generally present at abundances around 100 ppb; other elements are present at lower concentrations (Table 3).
Principle component analysis of these elemental concentrations, conducted with a varimax rotation, places five
elements (Ba, Ca, Na, Mg, Ni) on the first component (PC-1), with Ti, Rb, Si, K, Sr, and and Mn on the second (PC-
2). Together these two components explain 78% of the variance. Highest values of PC-1 are found in the Spring
samples, followed by the Stream and the two rock glaciers. In contrast, PC-2 is highest at the rock glacier sites and
lower in the Stream and Spring. Plotting of PC-1 vs. PC-2 reveals a nearly complete separation between the rock
glacier water and samples from the Stream and Spring (Figure 7). When considered as time-series, values of both
components are generally stable at the Stream and Spring, but rise consistently through the summer and fall at RG-1
and RG-2 (Figure 8).

The same 12 cations were generally detectable in the precipitation samples, with the exception of Fe, which
was typically below the detection limit. Values of Na, K, Mn, Rb, Fe, Ni, and Sr were higher in snow samples
relative to rain, with particularly high values of Na and K in the July snow sample from the RG-2 site (Figure 9). In
contrast, Ca, Ti, Ba, Mg, and Si were more abundant in rain samples. All elements were less abundant in rain than
in the time-series.

**5 Discussion**
**5.1 Isotopes and Hydrochemistry**
The automated samplers utilized in this project were successful at collecting essentially uninterrupted sequences of
water throughout their deployment. Modification of the samplers effectively reduced evaporation-related
fractionation that could have skewed the results over the long duration deployments. As seen in Figure 3, analysis
of the subsample from the first sample bottle that was removed in early July yielded similar results to the water that
remained inside the sampler until September. Values of $\delta D$ and $\delta^{18}O$ overlap within error for RG-2 and are very





| Table 3: Summary Hydochemistry | Na | K | Ca | Ti | Mn | Rb | Ba | Mg | Si | Fe | Ni | Sr |
|---|---|---|---|---|---|---|---|---|---|---|---|---|
| | (ppb) | (ppb) | (ppb) | (ppb) | (ppb) | (ppb) | (ppb) | (ppb) | (ppb) | (ppb) | (ppb) | (ppb) |
| **Stream (n=31)** | | | | | | | | | | | | |
| Mean | 848.2 | 465.8 | 1636.9 | 3.2 | 3.9 | 0.4 | 64.6 | 496.0 | 1053.2 | 66.9 | 0.8 | 9.0 |
| Median | 790.0 | 347.3 | 1635.9 | 3.2 | 2.4 | 0.3 | 61.6 | 499.7 | 1056.7 | 49.3 | 0.6 | 9.0 |
| Standard Deviation | 199.6 | 393.4 | 77.9 | 0.3 | 4.7 | 0.1 | 8.9 | 17.2 | 79.9 | 80.0 | 0.4 | 0.9 |
| Minimum | 605.4 | 262.6 | 1471.5 | 2.5 | 0.7 | 0.2 | 56.2 | 451.5 | 866.0 | 0.0 | 0.2 | 7.7 |
| Maximum | 1325.4 | 2264.8 | 1836.5 | 4.2 | 23.2 | 0.9 | 92.3 | 528.9 | 1176.2 | 410.1 | 1.6 | 11.2 |
| **Spring (n=33)** | | | | | | | | | | | | |
| Mean | 1075.5 | 499.9 | 1997.5 | 4.4 | 3.3 | 0.3 | 115.6 | 599.0 | 2034.7 | 34.7 | 0.7 | 11.0 |
| Median | 1104.8 | 513.1 | 1860.8 | 3.9 | 3.2 | 0.3 | 104.2 | 579.6 | 2048.8 | 20.1 | 0.5 | 10.8 |
| Standard Deviation | 284.8 | 107.3 | 313.0 | 0.9 | 1.6 | 0.1 | 34.9 | 56.8 | 84.4 | 33.6 | 0.6 | 1.1 |
| Minimum | 685.5 | 306.8 | 1764.4 | 3.6 | 1.1 | 0.2 | 90.7 | 517.4 | 1816.1 | 7.3 | 0.2 | 9.0 |
| Maximum | 2147.7 | 877.6 | 3100.8 | 7.5 | 6.7 | 0.7 | 251.1 | 770.6 | 2204.3 | 164.1 | 2.9 | 14.1 |
| **RG-1 (n=33)** | | | | | | | | | | | | |
| Mean | 619.8 | 612.0 | 1587.5 | 12.5 | 12.3 | 1.2 | 53.6 | 527.9 | 2265.6 | 277.3 | 0.7 | 12.3 |
| Median | 633.4 | 602.9 | 1550.2 | 11.5 | 4.3 | 1.1 | 52.6 | 506.7 | 2145.7 | 234.1 | 0.7 | 11.7 |
| Standard Deviation | 123.5 | 207.3 | 396.4 | 6.3 | 30.8 | 0.7 | 18.8 | 132.0 | 1008.7 | 215.0 | 0.4 | 4.3 |
| Minimum | 380.9 | 259.0 | 938.2 | 3.5 | 1.4 | 0.2 | 27.7 | 309.0 | 678.5 | 18.1 | 0.1 | 5.6 |
| Maximum | 898.9 | 1081.6 | 2527.2 | 25.8 | 173.6 | 2.6 | 120.5 | 830.7 | 4108.3 | 1041.1 | 1.8 | 20.8 |
| **RG-2 (n=34)** | | | | | | | | | | | | |
| Mean | 573.6 | 388.7 | 1289.7 | 6.1 | 8.1 | 0.7 | 43.3 | 396.9 | 1301.1 | 127.7 | 0.3 | 7.6 |
| Median | 570.9 | 372.2 | 1257.2 | 4.8 | 2.7 | 0.5 | 36.9 | 380.0 | 1225.0 | 56.3 | 0.2 | 7.0 |
| Standard Deviation | 170.6 | 95.0 | 257.4 | 4.1 | 14.2 | 0.4 | 17.3 | 87.3 | 444.9 | 174.2 | 0.3 | 2.7 |
| Minimum | 338.7 | 249.0 | 852.3 | 1.3 | 0.4 | 0.1 | 23.9 | 266.7 | 568.9 | 0.0 | 0.0 | 3.7 |
| Maximum | 883.8 | 592.8 | 1849.8 | 22.5 | 60.6 | 2.0 | 95.4 | 621.9 | 2323.0 | 764.2 | 1.8 | 15.1 |


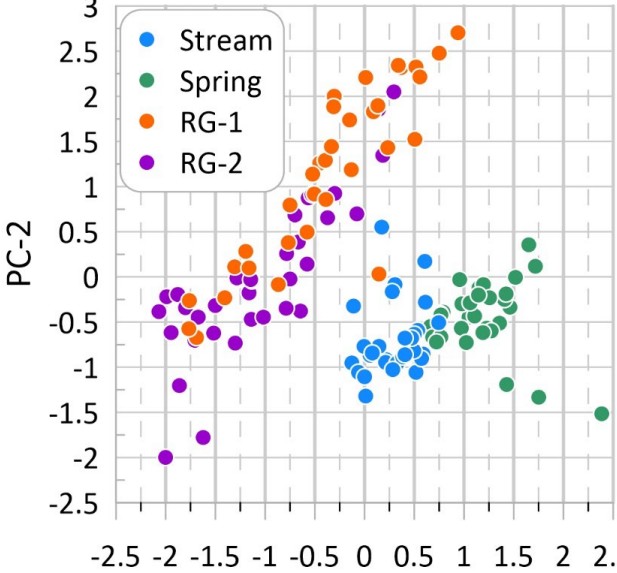

**Figure 7**: Biplot of the first and second principle components determined for major elements in the water samples. The similarity of the rock glacier water samples is clear, as is their lack of overlap with the Stream and Spring samples.



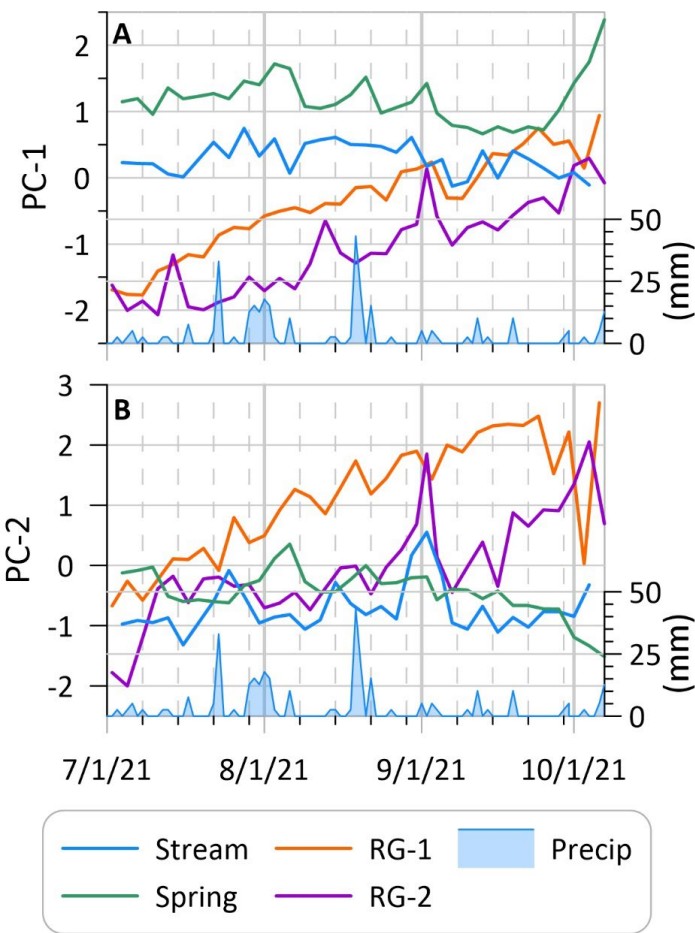

**Figure 8**: Time series of the first and second principle components presented in Figure 7. Values tend to be stable through the melt season at the Stream and Spring, but rise notably at both the rock glacier sites.

close for the Stream and Spring. There is a slight tendency for $\delta^{18}O$ to increase ~0.2‰ over the course of the summer, which is consistent with evaporation, however this shift is far less than the changes observed in these sequences of samples from start to finish. Thus, the time-series are interpreted without significant concern that values were altered by evaporation.

The sampling and analysis strategy in this project was designed to evaluate whether water draining from representative rock glaciers in the Uinta Mountains differs from streamwater and ground water in a manner that is consistent with the presence of melting ice within the rock glacier. The summer of 2021 was a particularly appropriate time to attempt this because the snowpack during the preceding winter was notably below average. On



April 1, 2021 the Chepeta SNOTEL (Figure 1) was at 83% of the 1991-2020 median of 380 mm snow water

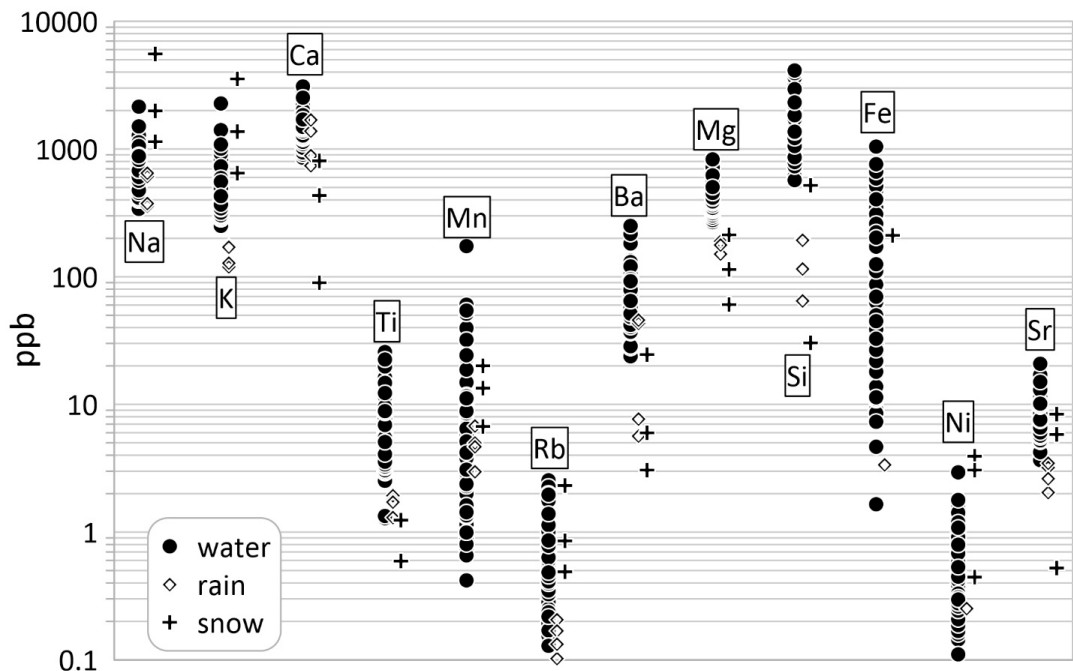


**Figure 9**: Abundances of detectable elements in the time series of water samples, along with the rain and snow
samples. Note the logarithmic scale on the Y-axis.

equivalent (SWE), but by April 13, the average date of the annual peak, SWE was just 52% of average (188 mm).
In contrast to other years in which we have conducted fieldwork at the RG-1 and RG-2 sites, the surfaces of the rock
glaciers were notably snow free when the samplers were deployed in early July. Inspection of high-resolution
satellite imagery confirms that visible snow on the rock glacier surfaces disappeared by the end of June, and that
essentially no snow was present anywhere in the West Fork Whiterocks Drainage after the start of July. Therefore,
it is unlikely that the water collected at RG-1 and RG-2, particularly late summer and fall, was sourced from melting
snow.

Analysis of stable isotopes reveals contrast between the water types that can be linked back to their sources

and flowpaths. Groundwater from the spring exhibits the most depleted $\delta^{18}O$, with values similar to snow (Figure
6). This correspondence suggests that the groundwater system is primarily recharged by snowmelt. The average
annual maximum SWE at the Chepeta SNOTEL station of 380 mm equals half of the mean annual precipitation.
The snowmelt pulse in the spring, therefore, is apparently the only precipitation event of the year that can
overwhelm the moisture holding capacity of the soil and pass water into the groundwater system. The consistently
low $\delta^{18}O$ values of the water discharging from the spring during the course of the summer, despite numerous
rainstorms delivering isotopically less depleted water, emphasizes that the deeper groundwater system is snow-
dominated and stable.



Samples from RG-1 and RG-2 from the early part of the melt season plot on the GMWL, with low values

consistent with a large contribution of snowmelt (Figure 5). Even though visible snow was absent from the rock

glacier surfaces at this time, this correspondence indicates that snow was still melting within the interstices between

blocks on the rock glacier surface, a situation that was reported by previous studies (Krainer et al., 2007). By the

beginning of August, however, isotope values at both rock glacier springs depart from the GMWL and rise to higher

values of *d-excess* (Figure 5). This pattern is not seen in the Spring or the Stream time-series, which remain on the

GMWL from start to finish. Thus, late summer and fall water discharging from both rock glaciers is distinct from

contemporary precipitation and ground water. This pattern is particularly dramatic at RG-1, where all of the August

through October samples cluster around a $\delta^{18}O$ of -13‰ with *d-excess* values as high as 20‰. Previous work on

rock glacier hydrology has reported high values of *d-excess* in late-summer rock glacier discharge, and interpreted

them as a signal of melting internal rock glacier ice that has undergone numerous freeze/thaw cycles (Steig et al.,

1998; Williams et al., 2006).

The time-series of PC values reinforce the uniqueness of the rock glacier water. Values for the Spring and

for the Stream are notably stable through the summer and into the fall (Figure 8). This consistency suggests that

these systems are not directly impacted by short-term events like rainstorms, or even changes over seasonal

timescales, presumably due to their well-mixed nature and large reservoirs. In contrast, the time series for the two

rock glacier springs rise dramatically during the melt season. Values of PC-1 increase starting at the beginning of

July in both records; values of PC-2 start rising in July for RG-1 and in mid-August for RG-2. Concentrations for

many individual elements increase by a factor of 3 or more from early July until October. This enrichment is

consistent with movement of water through the fine matrix of crushed rock material in the rock glacier interior,

where fresh mineral grains are available for rapid chemical weathering by cold water charged with carbonic acid

(Krainer and Mostler, 2002; Williams et al., 2006). Melting of ice would both liberate meltwater and open

flowpaths through this material. The pattern of rising dissolved load through the summer, therefore, provides

additional support for the interpretation that the source of the water draining from the rock glaciers shifts after

snowmelt is over.

The transition in source of the water draining from the rock glaciers is further illustrated by biplots of $\delta^{18}O$

against Ca and Mg content (Figure 10). Values for snow, rain, and the last sample from each rock glacier define a

triangle entirely enclosing samples collected from the rock glaciers. Water draining from the rock glaciers in July

exhibits a clear snowmelt influence, but this diminishes in August as the water becomes a more even mixture of rain

and rock glacier water. Through September into October, this balance shifts away from rain, eventually reaching a

minimal rain contribution in the last water discharged before the system froze up for the winter.

Williams et al. (2006) proposed a model for changing flowpaths and water sources over the course of the

melt season that is relevant for interpreting the results presented here. In early summer, the interior of a rock glacier

is frozen and water derived from snowmelt is discharged after draining through the blocky surface layer and running

along the top of the frozen core (Krainer and Mostler, 2002). Later in the summer, snowmelt is finished and



seasonal ice within the rock glacier begins to melt, opening flowpaths that bring meltwater into contact with fresh

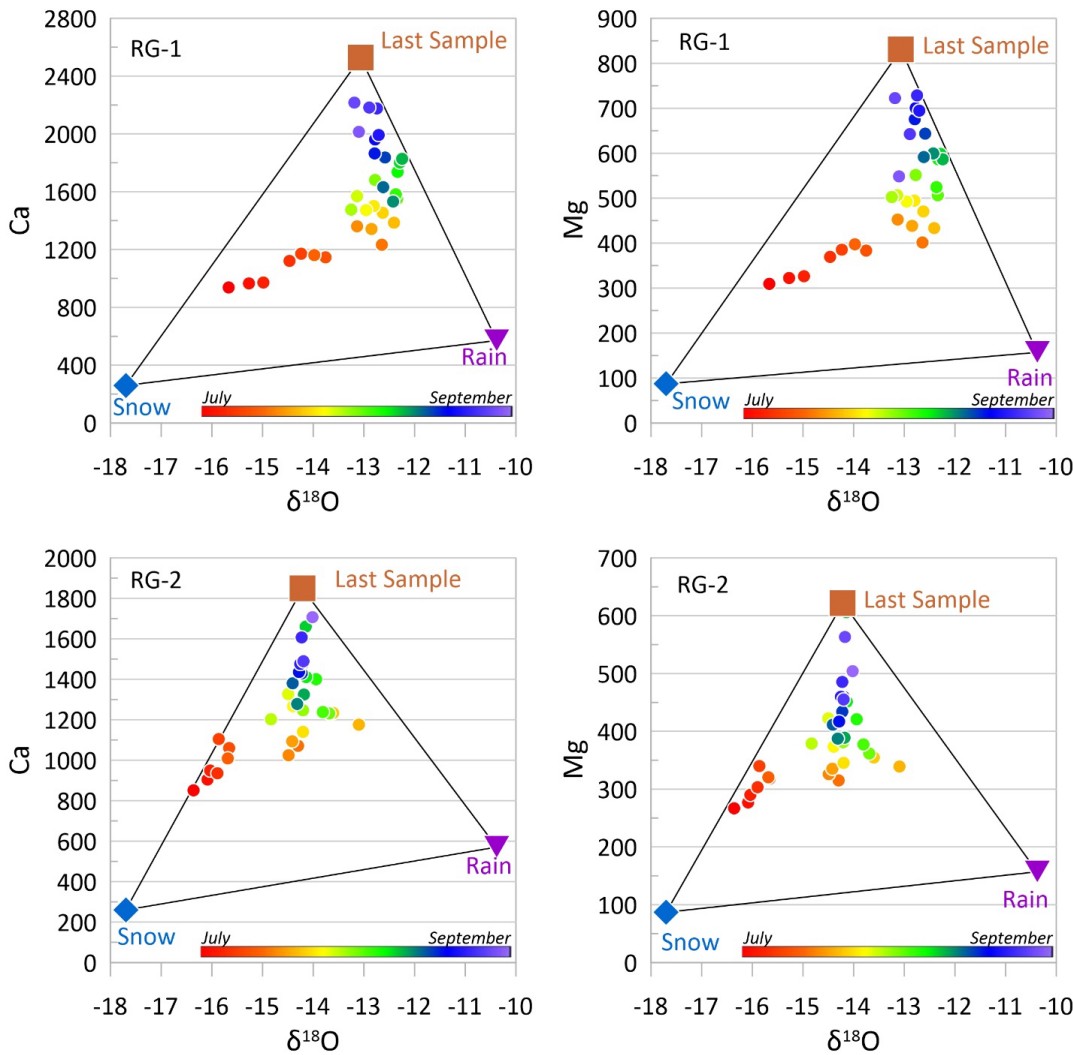

**Figure 10**: Biplots of $\delta^{18}O$ vs. Ca and Mg at RG-1 and RG-2. Circles represent water discharging from the rock glacier springs, with the rainbow pattern progressing from early July (red) through to early October (purple). The last sample collected at each rock glacier is plotted as the brown square, along with average values for snow and rain. Rock glacier water clearly evolves through the season from a composition dominated by snowmelt, to a mixture of rain and internal water, with decreasing rain influence over time.

highly weatherable mineral grains. Finally, in late summer and the fall, older perennial ice within the rock glacier
begins to melt (Williams et al., 2007), liberating water with high dissolved load and uniquely high values of *d-excess*
due reflecting a history of multiple freeze/thaw cycles. The isotopic and hydrochemical results presented here are
consistent with this model, supporting the interpretation that water discharging from Uinta rock glaciers in late
summer and fall is derived from the melting of perennial internal ice.

### 5.2 Implications for High Mountain Hydrology

The rock glaciers studied in this project are but two of eight mapped within the West Fork Whiterocks watershed
(Munroe, 2018), which also hosts extensive talus (Munroe and Laabs, 2009) that may contain non-trivial amounts of
ice. It is reasonable to predict, therefore, that water derived from rock glaciers may comprise an important amount
of the overall streamflow in the latter part of the summer and fall. Figure 11 presents biplots of $\delta^{18}$O vs. Ca and Mg
content, two elements that are notably elevated in the late summer rock glacier water in the Uintas and elsewhere
(Williams et al., 2006).

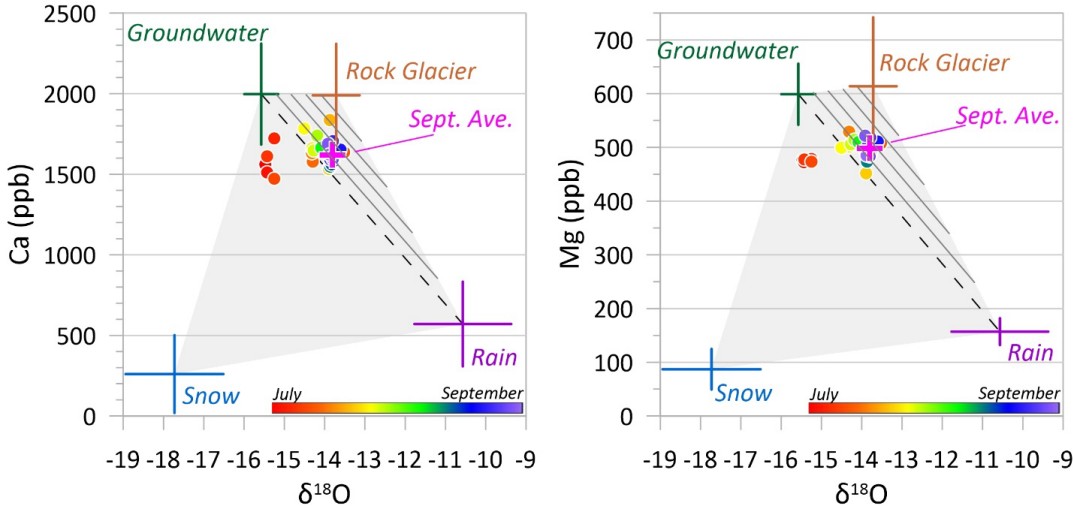

**Figure 11**: Biplots of $\delta^{18}$O vs. Ca and Mg used to determine the contribution of rock glacier discharge to streamflow. Water in the stream is plotted with a rainbow pattern progressing from July (red) through to early October (purple). Crosses represent the end members of snow, rain, groundwater, and rock glacier water. July streamwater samples contain a mixture of snowmelt and groundwater, but in August and September, snowmelt is now longer detectable (samples to right of dashed line). Streamwater samples at this time contain non-trivial amounts of water derived from rock glaciers, with an average of 25% in September (pink cross). Diagonal black lines denote the abundance of rock glacier water in increments of 20% for emphasis.

After Krainer and Mostler (2002), four end member sources of water to the stream are: groundwater, snow, rain, and
rock glaciers. The groundwater end member is constrained for the 2021 melt season by the 33 samples from the
non-rock glacier spring. The snow end member is less well constrained, however the two grab samples collected in
July from RG-1 and RG-2 have isotope values consistent with snow previously analyzed from the Uintas (Munroe,
2021) and with values predicted by the OPIC. Thus these samples are considered a valid representation of the snow
lingering in the Whiterocks River watershed in the summer of 2021. Five samples (two from RG-1 and three from
RG-2) collected from the rock glacier springs in October immediately before freeze up represent the rock glacier
meltwater end member. Finally, two precipitation samples from RG-2 and two from the spring site are available to



represent rain.  However,  the concentration of Ca in the Spring sampler is ~3x higher than at RG-2,  despite the
distance of only 3 km between the two sites. The precipitation sampler at the Spring site is located close to a dirt
road though, raising the possibility that dust produced by vehicle traffic raised the Ca content of the water collected
at this site.  Support for this interpretation is provided by 7 years of unpublished precipitation chemistry (n=79
samples) collected by the USDA-Ashley National Forest in the Uintas.  Concentrations of Ca in this dataset average
645 ppb, similar to the value of 570 ppb in the rain from the precipitation sampler at RG-2 and notably less than
mean of 1535 ppb at the roadside Spring site.  Thus, the precipitation samples from RG-2 alone are taken to
represent the rain end member in the stream system for the melt season of 2021.

With this approach, the four end members define a polygon entirely surrounding the streamwater samples

(Figure 11).  As in the time-series from the individual rock glaciers (Figure 5), a clear transition is notable. July
streamwater samples exhibit $\delta^{18}O$ values similar to snowmelt and groundwater, whereas late summer and fall
samples plot entirely within a triangle bounded by the groundwater, rock glacier, and rain.  Within this triangle,
although the proportions vary somewhat between samples, individual streamwater samples from August and
September can be visually separated as a mixture of ~20-30% rain, ~25 to 75% groundwater, and up to 50% rock
glacier water.  The overall mean of September streamwater samples can be defined as ~25% rain, ~50%
groundwater, and ~25% rock glacier water.  Water with a signature similar to that of springs discharging directly
from rock glacier termini, therefore, generally makes up a quarter of all the water flowing in the master stream of
this drainage after snowmelt has ended.

With such a significant contribution of rock glacier meltwater to streamflow in this system, it is worth

considering whether rock glacier ice is melting at an unsustainable rate. This possibility is hard to evaluate directly,
given that mass balance techniques for ice glacier systems are difficult to apply to rock glaciers (Østrem and
Brugman, 1966).  Nonetheless, it is notable that a depression consistent with subsidence accompanying the melt-out
of an ice core is present in the upper part of RG-2 (Figure 12).  A high-resolution topographic model constructed for
this rock glacier using structure-from-motion applied to images collected with an uncrewed aerial vehicle (UAV)
reveals that this depression has an area of 19,350 m$^2$ and a volume of 106,500 m$^3$ (mean depth of 5.5 m).  If this
depression formed due to the loss of ice, this volume corresponds to ~$10^8$ L of water.  At rates of $10^2$ to $10^3$ L/min
estimated for the modern flow, that equates to 70 to 700 days.  Thus, ice within this rock glacier may have begun
melting unsustainably in the past few decades in response to rising summer temperatures noted in Uinta climate
records (Brencher et al., 2021).  Future InSAR monitoring may help constrain subsidence on this and other rock
glaciers, yielding additional information about the response of these features to contemporary climate warming and
likely changes in their future contributions to high-elevation hydrology.

**6 Conclusion**

Time series of samples collected during the summer of 2021 reveal that water draining from rock glaciers in the
Uinta Mountains of Utah (USA) has a composition distinct from groundwater and from water in the master stream

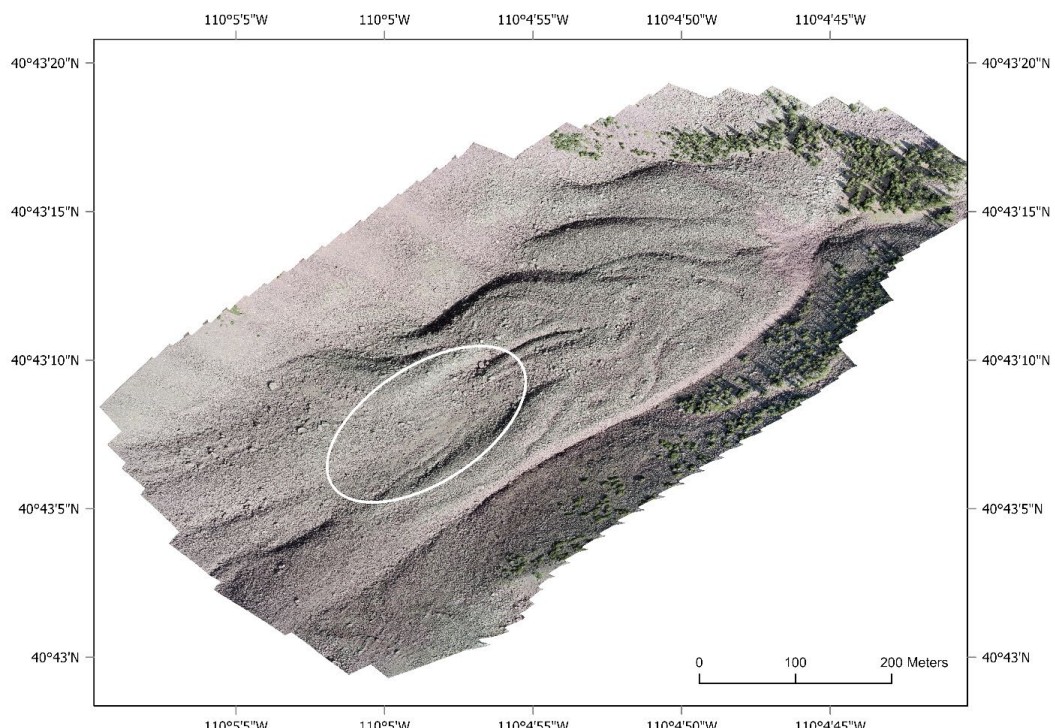

**Figure 12**: True-color hillshaded photomosaic of RG-2 produced by structure from motion (SfM) applied to a set of 243 images collected at an altitude of 120 m above the ground. The while oval highlights the prominent depression near the head of the rock glacier, which may reflect subsidence due to ice meltout.

of a representative 5000-ha drainage. Rock glacier water resembles snowmelt in the early summer, but transitions to higher values of *d-excess* and greatly elevated Ca and Mg content as the melt season progresses. This pattern is consistent with models describing a change in water source from snowmelt, to melting of seasonal ice, to melting of deeper perennial ice in the rock glacier interior in late summer and fall. Water derived from this internal ice appears to have been the source of ~25% of the streamflow in this study area during September of 2021. This result emphasizes the significant role that rock glaciers can play in the hydrology of high-elevation watersheds, particularly in melt seasons following a winter with below average snowpack.





**Author Contributions**
JM designed the project, conducted the fieldwork and laboratory analyses, interpreted the results, and drafted the
figures. JS prepared the manuscript with contributions from AH.

**Competing Interests**: The authors declare that they have no conflict of interest.

**Acknowledgements**
This work was supported by NSF HS-1935200 to PIs Munroe and Handwerger, and NSF MRI-1918436 to Munroe.
The authors thank Q. Brencher, C. Kluetmeier, S. Lusk, E. Norris, A. Santis, and A. Takoudes for their assistance in
the field, and E. McMahon for help preparing the water samplers.



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
