# Peer review of "Contribution of Rock Glacier Discharge to Late-Summer and Fall Streamflow in the Uinta Mountains, Utah, USA"

_Hydrology and Earth System Sciences, 2022_

## Referee Comment (RC1)

Comments on the manuscript

**Contribution of Rock Glacier Discharge to Late-Summer and Fall Streamflow in the Uinta Mountains, Utah, USA**

by Jeffey S. Munroe and Alexander L. Handwerger

The authors analyzed water released from two active rock glaciers in the Uinta Mountains of Utha (USA) which they compare with water released from a spring that is not related ro rock glaciers, and with water of the master stream of the catchment area.

Water samples ware taken by automtaic water samplers from the beginning of July until mid October in 2021 for analyizing stable isotopes and hyrochemistry (ICP-MS). Additional samples were taken from the snowpack and also rainwater was collected.

The authors conclude that water released from the rock glaciers is derived from snowmelt in late summer shifting to seasonal ice and melting of perennial ice in fall.

The manuscript is an important contribution to the understanding of the hydrological/hydrogeological system of intact (active) rock glaciers, mainly based on the study of stable isotpes and hydrochemistry.

**Some general comments**
A great disadvantage of the study is the short sampling period which started at the beginning of July and therefore did not cover the beginning and peak of the snowmelt period which would be important for understanding the hydrological and hydrochemical processes. Another disadvantage is the lack of continuous records of the water height (discharge) and electrical conductivity. It is well known that discharge of intact rock glaciers is characterized by distinct seasonal and diurnal variations, particularly during the snowmelt period.
To see the influence of individual rainfall events on isotope values and hydrochemistry a much shorter sampling interval (one hour) is needed.
Therefore I suggest to continue the hydrological studies on these rock glaciers and to extend the sampling/measuring period to include the beginning of the melt season and also to install a

gaging station that records water height and electrical conductivity. This will support the interpretation of isotope and hydrochemical data.

An average discharge of 15 L/min seems to be extremely low, too low, even when the annual precipitation rate is lower than in the Alps. So may be that a major portion of the discharge is released as groundwater stream and not as surface stream?

Tracer tests (e.g. dye tracer test) will provide useful information on the residence time (flow velocity) of the quickflow component (includes water derived from snowmelt, rainfall events and partly also from melting of ice) which for rock glaciers of medium-size is in the order of a few hours.

The presence of a massive ice core indicates that below the frozen core a layer of unfrozen, fine-grained sediment may occur which acts as a reservoir for groundwater that is supplied by snowmelt, rainfall and icemelt, and slowly flows through the reservoir and is characterized by high electrical conductivity.

Geophysical investigations would provide information on the internal structure of the rock glacier (thickness of the  active layer, thickness and composition of the frozen core, presence of an unfrozen sediment layer between the bedrock and the frozen core).

The title clearly reflects the contents of the manuscript

The abstract provides a brief and informative summary

The overall presentation is well structured and clear

The aim of this study is clearly pointed out, the manuscript addresses relevant scientific questions and the authors present new ideas and data concerning the hydrology of intact rock glaciers

Study area is precisely described, the scientific methods are valid and clearly outlined

The results are sufficient to support the interpretation and conclusions (see comments). The results are well presented, the text is supported by a number of instructive figures

**Conclusions**

The authors conclude that in early summer discharge is mainly composed of water derived from snowmelt.

Additionally, rainfall events also contribute to the discharge (indicated by rainfall events shown in Fig. 8). The proportion of rainwater is low in early summer and icreases towards fall (untill refreezing), this is not considered in the conclusions. The proportion of icemelt will also be very low in early summer and increase towards fall as concluded by the authors. There will also be a small amount of groundwater that is not mentioned in the conclusions. The proportion of groundwater will be low in early summer and increase towards fall. There is probably also groundwater released during the winter months.

Figures and Tables are clear and instructive

The number and quality of the references is appropriate. The authors give proper credit to realted work and clearly indicate their own contributions. The reference list is complete

See additional comments in the manuscript

---

## Referee Comment (RC2)

Review: **Contribution of Rock Glacier Discharge to Late-Summer and Fall Streamflow in the Uinta Mountains, Utah, USA**

Manuscript number: hess-2022-285

Summary: This manuscript presents a field study designed to investigate the contribution of rock glaciers to the streamflow regime of a 50 km2 mountainous watershed in the Uinta mountain range, in the Colorado Rockies. Water samples were collected over a summer-fall season, by means of automated samplers located at the outlet of two rock glaciers, a groundwater spring, and the stream channel. Additional samples of snow (grab), snowmelt and rain were obtained. All samples were analyzed for stable water isotopes and for dissolved solids. Through a diverse set of graphical analyses, plus an end-member mixing method, the manuscript makes a case for a conceptual model of sources contributions to streamflow and attempts to quantify the importance of rock glacier ice melt to river flow, which amounts to 25% according to the author's estimate during this summer season.

General comments: this is a well written and nicely presented work, which has the potential of becoming a welcome addition to the body of literature. The study area, methods and results are presented, by and large, in a systematic and coherent way. The figures are of good quality, although the look of the tables could be improved. In terms of the analysis, this paper is affected by a common problem, which is to properly characterize the chemical and isotopic composition of seasonal snow. The number of snow samples is very limited, which is understandable given the difficulties of carrying out field work in mountainous areas. However, there's documented evidence of the spatial and temporal variability of snow (and, importantly, snowmelt) isotopic composition. This variability can hamper the possibility of portraying a clear-cut picture of streamflow sources, but the authors neglect to explicitly discuss this source of uncertainty. I suggest they review, for example, the following references:

Cable J, Ogle K, Williams D. 2011. Contribution of glacier meltwater to streamflow in the Wind River Range, Wyoming, inferred via a Bayesian mixing model applied to isotopic measurements. Hydrol. Process. 25 (14): 2228–2236. DOI:10.1002/hyp.7982

Ladouche B, Probst A, Viville D, Idir S, Baqué D, Loubet M, Probst J-L, Bariac T. 2001. Hydrograph separation using isotopic, chemical and hydrological approaches (Strengbach catchment, France). J. Hydrol. 242(3–4): 255–274. DOI:10.1016/S0022-1694(00)00391-7

Laudon H, Slaymaker O. 1997. Hydrograph separation using stable isotopes, silica and electrical conductivity: an alpine example. J. Hydrol. 201(1-4): 82–101. DOI:10.1016/S0022-1694(97)00030-9

Rodriguez, M., Ohlanders, N., Pellicciotti, F., Williams, M.W. and McPhee, J., 2016. Estimating runoff from a glacierized catchment using natural tracers in the semi-arid Andes cordillera. *Hydrological processes*, *30*(20), pp.3609-3626.

The discussion section includes an estimation of % contribution to streamflow, and an analysis of the degree of imbalance of the rock glaciers with current climate. It is argued that observed subsidence could be a result of 1 to 10 years of ice melt, given rock glacier outlet flow estimates. Although this is an intriguing concept, too little information is provided to sustain this calculation, and it feels as if it was added at the last minute to the manuscript. I suggest fleshing this analysis out or deleting it altogether.

Specific Comments:

L77. Probably you mean that no *clean glaciers*, remain in the Uintas.

Fig1. This figure could be complemented with a layer showing the delineation of known rock glaciers in the basin. Is the Spring sampler upstream of the Stream sampler? it is not clear from the figure.

L90. Replace "samples" with "samplers".

L107-109. Were these data loggers active for this research? if so, their location should be shown in Figure 1.

L115. This range of flow is 1 to 2 orders of magnitude larger than the discharge from RG-1. Why might this be? Before, you say that both glaciers are "600 m long and 100 m wide". Can you be more specific and provide surface area estimates for each rock glacier?

Methods section. Please provide accuracy estimates for isotopic signatures based on the analytical procedures and instruments employed.

L151. Replace "was" with "were".

Figure 3. Why are samples from RG1 not shown in this figure? I can't see a mention to this in the text.

Table 2. RG1 samples seem to be less depleted than those from RG2. Is this difference significant?

Figure 4. Why are snow, rain and melt samples not plotted in this graph? the number of samples in these cases is very low, but nonetheless it would be interesting to see where they fall in the graph.

Figure 5. Why are symbols for RG2 in July different?

Figure 6. It makes little sense to plot snow values as an average, with so few samples. Better just plot the individual samples in the graph. Same with snowmelt.
L250. The technique is called "principal" component analysis. Please review and correct throughout the text.

Figure 9. It would be nice to try and discriminate the water samples among stream, spring and rock glaciers. Maybe you could select major ions and plot in this graph as well.

L311. This inference might be correct, but I think that it is unsupported by the available data, which is very scarce in terms of snow and snowmelt isotopic composition.

Snow at different altitudes can have a large spread in the isotopic signal, and melt can favor preferential elution, which muddles the picture when trying to link stream and snow samples.

L318 and elsewhere: please use a more direct time referencing to help the reader follow your analysis. Talk in terms of specific months, at the very least.

L318. low values of what?

L333. these large reservoirs have not been described previously in the text (except for one lake). They should be mentioned in the study area description and their storage volume at least approximately quantified.

Figure 10. In this end-member mixing analysis, error bars should be provided, moreover considering the very low number of snow samples.

L384. "Thus these samples…" Yes, but may not necessarily be a valid representation of snow MELT at this time, or of basin-wide snow composition, because of preferential elution and spatial (elevation-dependent) isotopic signatures of accumulated snow. The authors must discuss this source of uncertainty and incorporate somehow in their estimates.

L389-395. the fact that this potential bias in the data (originated by the road) is brought up this late in the paper seems problematic to me. Are these data included in all the previous analysis? Why? should this data be discarded altogether?

L403. The authors should discuss why this significant estimated contribution from RG is not reflected in the stream isotopic composition time series, which remains stable although there's a strong trend in the RG series (figure 6).

L413. but you also estimated 15 l/min for one of the glaciers. So, the rate of ice melt is hugely uncertain! Although this surface depression analysis is interesting, the way it is presented here feels rushed and somewhat contrived. The authors should expand: what is the total area of RG mapped in this basin? what is the range of glacier ice available? etc.

Figure 12. I'm afraid that I can't see the depression the authors refer to. This data should be presented much earlier, in the data section, and not in the discussion section.

L428. Based on only one sample of snowmelt water, it is tenuous to make strong statements about snowmelt similarity or influence on streamflow throughout the season. This is a major problem of the material presented here, and should be discussed by the authors.

---

## Author Comment (AC1)

**Response to Referee #1**

We appreciate the thoughtful suggestions of Referee #1 and their supportive assertion that our "*manuscript is an important contribution to the understanding of the hydrological/hydrogeological system of intact (active) rock glaciers*". Our goal in this study was to evaluate the contribution of water from rock glaciers to the overall hydrology of a high-elevation watershed, particularly late in the melt season. We are happy to see that the reviewer felt that we successfully achieved this objective.

Nonetheless, the Referee did raise several points in their general comments to which we wish to respond here:

First, the Referee felt that our sampling period was too short and that it was a disadvantage that we did not sample the early phase of snowmelt. We offer two arguments in response. The first is practical: the sampling site is more than 30 km away from the nearest plowed road and is completely inaccessible from the start of winter through late June. Thus it was not possible to collect water during the early summer. More significantly, the focus of our study was on the composition of water late in the melt season, thus collecting snowmelt-dominated water early in the year was unnecessary. We admit that in a perfect world it could be illuminating to collect water continuously from the onset of melt through freeze-up in the fall. Yet, given the snow-dominated nature of this mountain system, we would predict that the water we failed to collect in May/June would be nearly 100% snowmelt and not germane to our study objectives.

Second, the Referee notes that it could have been useful for us to collect measurements of spring discharge and electric conductivity. We agree, but unfortunately neither of these was possible during the summer of 2021 when our water samples were collected. Installation of a weir for recording discharge was not allowed in the terms of our research permit, and previous studies (as well as the reviewer themself) have noted the difficulty in accurately assessing rock glacier spring discharge because so much of the water flows below the ground surface. With respect to electric conductivity, a datalogger recording EC was deployed at one of the sites in this study (RG-1) during the summer of 2022. It recorded values that are low (8-16 µS/cm), but consistent with water draining from quartzite bedrock. Notably, values of EC rose steadily during the 2022 melt season, supporting the conclusion that late-summer discharge from this spring is dominated by water with a longer residence time and more extensive contact with fresh weatherable minerals.

Third, the Referee notes that a shorter sampling interval is necessary to see the influence of individual rainfall events. We argue that individual rainfall events, at least the more voluminous ones, are still detectable in our samples, which were collected at 12-hour intervals. As shown in our Figure 6, the local weather data to which we compare our data were collected at 1-hr intervals, and the major spikes in our data (hydrochemistry, isotopes) each align with precipitation events (or clusters of closely spaced storms).

Fourth, the Referee suggests that dye tracer tests would be helpful in evaluating the velocity of the quickflow component in these rock glacier systems. We agree, and although this was beyond the scope of our project, we will consider incorporating this approach in our future work.

Finally, the Referee mentions that a layer of unfrozen sediment could function as a groundwater reservoir below the implied frozen core of the rock glacier, and suggests that geophysical investigations could be used to evaluate this. We are aware of previous published models for rock glacial structure that postulate

such an unfrozen zone, and we would be interested in applying electrical resistivity or other techniques in the future to evaluate whether that model is valid for the rock glaciers we studied. As the Referee notes, such an unfrozen layer could yield water with higher EC, but the shifting isotope values and d-excess that we observed seem better explained by contributions from a reservoir of ice that has undergone numerous melt/freeze cycles.

We wish to thank the Referee for their careful reading of our manuscript, their encouraging assessment of our project's value, and their helpful suggestions of additional approaches to consider. Although we cannot go back to the summer of 2021 to make some of the measurements they recommend, we will certainly keep these suggestions in mind as we design future expansions of our rock glacier research.

---

## Author Comment (AC2)

**Response to Referee #2**

We appreciate the supportive tone and considerate comments that Referee #2 provided on our manuscript. The Referee's opinion that the manuscript is "*well written and nicely presented [...], which has the potential of becoming a welcome addition to the body of literature*" is encouraging and the list of specific comments will be very helpful in improving the overall presentation and interpretation of our data.

The Referee's overarching general comment is that our sampling strategy relying on a small number of grab samples may not adequately capture the isotopic and chemical variability in the seasonal snowpack, as has been well established by previous field studies. We appreciate that the Referee acknowledges the reality of the inherent challenges in collecting large numbers of snow samples in a mountainous field area, and we certainly wish we could have collected more and from earlier in the season. However, such an expanded sampling campaign was simply not possible. Thus, we are left with the samples we have and need to rely upon them, with their imperfections, as the sole source of information about the composition of the snow contributing meltwater to this system in 2021. In our revisions we will endeavor to more explicitly address this as an understandable but unavoidable limitation of our study, and to explore ways in which this limitation might carry over into our interpretations. The references suggested by the Referee will be very helpful in expanding this part of our paper. Ultimately, we are confident that our general conclusions are robust and are not substantially undermined by the uncertainty driven by the small number of snow samples. But we agree that we should do more to explicitly explore this for our readers.

The Referee also notes that our calculations comparing the subsidence volume on RG-2 with the estimated spring discharge seems like a "last minute" addition to the paper. However, we intentionally bring this in at the end of the discussion because our numbers are inherently imprecise (both the timing of when the observed depression formed and the true rate of spring discharge cannot be known). Thus, we are merely endeavoring to note that there is evidence of subsidence and that the magnitude of that subsidence is compatible with relatively recent change in the mass balance of ice in this rock glacier system. If we had more robust measurements, we could make this a more central part of the discussion. But given the uncertainties, it seems prudent to include this as more of a supporting note, rather than a main argument. Nonetheless, in our revisions, we will explore how to make the transition to this section smoother so that it doesn't seem like an afterthought.

Specific Comments (*with our response in italics*)
L77. Probably you mean that no *clean glaciers*, remain in the Uintas.
   *-The best way to distinguish "ice glaciers" from "rock glaciers" with minimal wordiness is elusive. Here we meant "ice glaciers"; perhaps "clean glaciers" is a better way to phrase that.*

Fig1. This figure could be complemented with a layer showing the delineation of known rock glaciers in the basin. Is the Spring sampler upstream of the Stream sampler? it is not clear from the figure.

*-We can easily add the mapped rock glacier outlines to the figure, and will clarify the relationship between the different samplers. Thank you for the suggestions.*

L90. Replace "samples" with "samplers".

*-We will make this change.*

L107-109. Were these data loggers active for this research? if so, their location should be shown in Figure 1.

*-The data from the loggers were presented in the referenced paper (Munroe, 2018). They were located at the same positions at the water samplers in this study.*

L115. This range of flow is 1 to 2 orders of magnitude larger than the discharge from RG-1. Why might this be? Before, you say that both glaciers are "600 m long and 100 m wide". Can you be more specific and provide surface area estimates for each rock glacier?

*-Flow estimates from rock glaciers are notoriously difficult because so much of the water can drain below ground, flowing through the rocks. We suspect that is the explanation for the difference between the estimates for these two springs. The rock glaciers are about the same size, but we can certainly add more specific dimensions including areas.*

Methods section. Please provide accuracy estimates for isotopic signatures based on the analytical procedures and instruments employed.

*-We will add these estimates in our revisions.*

L151. Replace "was" with "were".

*-We will make this change.*

Figure 3. Why are samples from RG1 not shown in this figure? I can't see a mention to this in the text.

*-It was not possible to return to the more distant RG-1 sampler early in the season to collect a subsample as we done with the other more accessible samplers. A note explaining this will be added to the text.*

Table 2. RG1 samples seem to be less depleted than those from RG2. Is this difference significant?

*-The difference is statistically significant, we will note this in our revisions.*

Figure 4. Why are snow, rain and melt samples not plotted in this graph? the number of samples in these cases is very low, but nonetheless it would be interesting to see where they fall in the graph.

*-We omitted snow, rain and melt samples in Figure 4 for clarity, but we presented their ranges in Figure 6. We will explore whether it is possible to present them in both places, but want to avoid compromising clarity of the figure.*

Figure 5. Why are symbols for RG2 in July different?

*-As noted in the figure caption, the small green diamonds present reconnaissance data collected for RG-2 in the fall of 2020 during a preliminary phase of this project.*

Figure 6. It makes little sense to plot snow values as an average, with so few samples. Better just plot the individual samples in the graph. Same with snowmelt.

*-Our logic in presenting the range was to acknowledge that our understanding of the exact snowmelt values is imperfect; presenting the range was a visual way to highlight this uncertainty. We can explore whether presenting the individual values provides the same effect.*

L250. The technique is called "principal" component analysis. Please review and correct throughout the text.

*-Our mistake, we will make this change throughout the text.*

Figure 9. It would be nice to try and discriminate the water samples among stream, spring and rock glaciers. Maybe you could select major ions and plot in this graph as well.

*-Good suggestion, we will explore ways to discriminate between the water samples with color and/or symbols.*

L311. This inference might be correct, but I think that it is unsupported by the available data, which is very scarce in terms of snow and snowmelt isotopic composition.

*-True, our number of snow samples is small. But here we are noting the correspondence between the similarly depleted isotope values for both snow and groundwater, and emphasizing that groundwater values are most depleted. The most logical explanation for that observation is that groundwater is primarily derived from snowmelt.*

Snow at different altitudes can have a large spread in the isotopic signal, and melt can favor preferential elution, which muddles the picture when trying to link stream and snow samples.

*-True, we will be sure to include mention of this in the revisions we make.*

L318 and elsewhere: please use a more direct time referencing to help the reader follow your analysis. Talk in terms of specific months, at the very least.

*-Thank you for the suggestion, we will endeavor to do so in our revisions.*

L318. low values of what?

*-As we are referring to the GMWL here, we thought it would be clear that we are talking about low value of deuterium and $\delta^{18}O$. We will make this clearer in our revision.*

L333. these large reservoirs have not been described previously in the text (except for one lake). They should be mentioned in the study area description and their storage volume at least approximately quantified.

*-We apologize if that was unclear; we are not referring to artificial water reservoirs,*

*rather we are noting that the spring and the stream systems are larger and more voluminous than the two rock glaciers, thus they would be more likely to exhibit stable values during the melt season.*

Figure 10. In this end-member mixing analysis, error bars should be provided, moreover considering the very low number of snow samples.

> *-We will explore how to include this information in our revision.*

L384. "Thus these samples…" Yes, but may not necessarily be a valid representation of snow MELT at this time, or of basin-wide snow composition, because of preferential elution and spatial (elevation-dependent) isotopic signatures of accumulated snow. The authors must discuss this source of uncertainty and incorporate somehow in their estimates.

> *-We will include a more robust exploration of this uncertainty in our revision. Nonetheless, the fact that our small number of samples yields isotope values similar to other collections from the region and to values predicted by the OIPC supports the assessment that our samples are not terribly influenced by post-depositional effects.*

L389-395. the fact that this potential bias in the data (originated by the road) is brought up this late in the paper seems problematic to me. Are these data included in all the previous analysis? Why? should this data be discarded altogether?

> *-We agree that the possibility that road dust influenced the precipitation samples collected at the spring site is concerning. This is why we relied on the precipitation collected at RG-2 (far from any road) to constrain precipitation chemistry (Lines 394-395).*

L403. The authors should discuss why this significant estimated contribution from RG is not reflected in the stream isotopic composition time series, which remains stable although there's a strong trend in the RG series (figure 6).

> *-In reality, only the spring (groundwater) time series is stable (Figure 6); the streamwater does exhibit rising isotope values late in the season that are consistent with an increased rock glacier component.*

L413. but you also estimated 15 l/min for one of the glaciers. So, the rate of ice melt is hugely uncertain! Although this surface depression analysis is interesting, the way it is presented here feels rushed and somewhat contrived. The authors should expand: what is the total area of RG mapped in this basin? what is the range of glacier ice available? etc.

> *-We previously discussed our logic in presenting the implications of this surface depression so late in the discussion. We will work to make sure this doesn't seem like an afterthought, while simultaneously balancing the reality that are calculations are imprecise. We could certainly add mention of the total area of rock glaciers mapped in the basin, and we previously mention that glaciers are absent in this region.*

Figure 12. I'm afraid that I can't see the depression the authors refer to. This data should be presented much earlier, in the data section, and not in the discussion section.

*-We agree that the depression is difficult for some to see given the direction of the apparent illumination in the surface model underlying the photo mosaic. We can adjust this in our revisions. As we have discussed already, given the uncertainties we feel it is prudent and appropriate to include description of this depression and its possible significance late in the discussion section. We will adjust the presentation though to make this transition smoother.*

L428. Based on only one sample of snowmelt water, it is tenuous to make strong statements about snowmelt similarity or influence on streamflow throughout the season. This is a major problem of the material presented here, and should be discussed by the authors.

*-As noted above, we will be certain to include a more explicit consideration of the limitations imposed by the small number of snow samples. At the same point, we obviously cannot go back in time to collect more snow and the Referee acknowledges the inherent difficult in making large-scale snow collections in such terrain. We are encouraged that the values for the samples we do have match so well with other reported isotopic values in snow from these mountains, and with the values predicted by the OIPC algorithm. We consider both of these as solid lines of support for the snow values we were able to directly measure.*

---

## Author Response (AR1)

We appreciate the opportunity to revise this manuscript in response to the thoughtful suggestions of the two reviewers. Here we present our responses to these suggestions.

Referee #1 felt that it was a disadvantage that we did not sample the early phase of snowmelt, but as we noted in our response document, the study area is inaccessible during winter due to deep snowpack. We added mention of this reality to the text (Lines 131-132)

Referee #2's overarching general comment is that our sampling strategy for snow may not adequately capture the isotopic and chemical variability in the seasonal snowpack. As noted in our response document, we certainly wish we could have collected more snow and from earlier in the season. However, such an expanded sampling campaign was simply not possible. Thus, we are left with the samples we have and need to rely upon them, with their imperfections, as the sole source of information about the composition of the snow contributing meltwater to this system in 2021. In our revised text we acknowledge that previous work has clearly established how isotope values in snow can vary spatially and change with time. However, we also emphasize that our measurements are consistent with other analyses of snow from the Uinta Mountains and with values predicted by the Online Isotopes in Precipitation Calculator. Thus, it seems appropriate to consider our values to be broadly representative of snow in the Uintas, which is sufficient for our purposes in this study (Lines 253-259).

The Referee also felt that our calculations comparing the subsidence volume on RG-2 with the estimated spring discharge seems like a "last minute" addition to the paper. However, as we explained in our response document, we intentionally placed this at the end of the discussion because our numbers are inherently imprecise (both the timing of when the observed depression formed and the true rate of spring discharge cannot be known). Thus, we are simply endeavoring to note that there is evidence of subsidence and that the magnitude of that subsidence is compatible with relatively recent change in the mass balance of ice in this rock glacier system. If we had more robust measurements, we could make this a more central part of the discussion. But given the uncertainties, it seems prudent to include this as more of a supporting note, rather than a main argument.

Specific Comments (*with our response in italics*)
L77. Probably you mean that no *clean glaciers*, remain in the Uintas.

*-Changed to "ice glaciers" (Line 77)*

Fig1. This figure could be complemented with a layer showing the delineation of known rock glaciers in the basin. Is the Spring sampler upstream of the Stream sampler? it is not clear from the figure.
*-We added the mapped rock glacier outlines to the figure, and clarified the position of the spring and stream samplers in the text. (Line 122)*

L90. Replace "samples" with "samplers".
*-We made this change. (Line 101)*

L107-109. Were these data loggers active for this research? if so, their location should be shown in Figure 1.
*-The data from the loggers were presented in the referenced paper (Munroe, 2018). They were located at the same positions at the water samplers in this study.*

Methods section. Please provide accuracy estimates for isotopic signatures based on the analytical procedures and instruments employed.
*-We added these statistics in our revisions. (Lines 146-148)*

L151. Replace "was" with "were".
*-We made this change. (Line 152)*

Figure 3. Why are samples from RG1 not shown in this figure? I can't see a mention to this in the text.
*-It was not possible to return to the more distant RG-1 sampler early in the season to collect a subsample as we done with the other more accessible samplers. A note explaining this was added to the text. (Line 134)*

Table 2. RG1 samples seem to be less depleted than those from RG2. Is this difference significant?
*-The difference is statistically significant, we noted this in our revisions. (Lines 190-191)*

Figure 4. Why are snow, rain and melt samples not plotted in this graph? the number of samples in these cases is very low, but nonetheless it would be interesting to see where they fall in the graph.
*-We experimented with presenting the precipitation values in Figure 4, but with so many data points already plotted for the water samples, this addition made the figures too crowded. The low values for the snow, in particular, also required adjusting the axes with the result that the datapoints for water samples were even more clustered. Because the intent with this figure is to visually present how the isotopic composition of the water samples evolves during the melt season, we elected not to include the precipitation values and reduce the readability. As noted below, ranges for the precipitation values are presented in Figure 6.*

Figure 5. Why are symbols for RG2 in July different?
*-As noted in the caption for Figure 5, the small green diamonds present reconnaissance data collected for RG-2 in the fall of 2020 during a preliminary phase of this project.*

Figure 6. It makes little sense to plot snow values as an average, with so few samples. Better just plot the individual samples in the graph. Same with snowmelt.
*-Presenting the range is a visual way to highlight that our snow measurements are but a few*

*datapoints in a spread of values. Plotting the individual points cluttered the figures and de-emphasized the likely continuum of snow values. We retained the brackets presenting the range.*

L250. The technique is called "principal" component analysis. Please review and correct throughout the text.
*-Our mistake, we made this change throughout the text.*

Figure 9. It would be nice to try and discriminate the water samples among stream, spring and rock glaciers. Maybe you could select major ions and plot in this graph as well.

*-We updated Figure 9 so that the water samples are presented with the same colors as in previous figures. We also note in the caption the order in which the four sites are presented from left to right to aid readers who do not have access to the color version.*

L311. This inference might be correct, but I think that it is unsupported by the available data, which is very scarce in terms of snow and snowmelt isotopic composition.
*-True, our number of snow samples is small. But here we are noting the correspondence between the similarly depleted isotope values for both snow and groundwater, and emphasizing that groundwater values are most depleted. The most logical explanation for that observation is that groundwater is primarily derived from snowmelt.*

Snow at different altitudes can have a large spread in the isotopic signal, and melt can favor preferential elution, which muddles the picture when trying to link stream and snow samples.
*-We added mention of this in our revisions. (Lines 252-259)*

L318 and elsewhere: please use a more direct time referencing to help the reader follow your analysis. Talk in terms of specific months, at the very least.
*-Thank you for the suggestion, we endeavored to do this throughout our revisions.*

L318. low values of what?
*-As we are referring to the GMWL here, we thought it would be clear that we are talking about low value of deuterium and $\delta^{18}O$. We made this more explicit. (Line 334)*

L333. these large reservoirs have not been described previously in the text (except for one lake). They should be mentioned in the study area description and their storage volume at least approximately quantified.
*-We changed "reservoirs" to "volumes". (Line 350)*

Figure 10. In this end-member mixing analysis, error bars should be provided, moreover considering the very low number of snow samples.
*-We considered how to include this information in our revision, but ultimately decided that it unnecessary. We are not, in Figure 10, attempting to quantify the abundance of each end*

*member in a given water sample. Rather, we are aiming to visually present evidence that the overall composition of the water samples collected at the two rock glacier sites during the coruse of the melt season transitions away from the typical composition of the snow (isotopically depleted and fresh) toward rain (less depleted and still fresh) toward something else (intermediate isotope values and much higher dissolved load). In Figure 6 and in the text we are clear that the values in the snow samples vary, but for the purpose of this visual argument, the range of the error bars is non-essential.*

L384. "Thus these samples…" Yes, but may not necessarily be a valid representation of snow MELT at this time, or of basin-wide snow composition, because of preferential elution and spatial (elevation-dependent) isotopic signatures of accumulated snow. The authors must discuss this source of uncertainty and incorporate somehow in their estimates.
*-We understand the reviewer's point, but would argue that the similarity of our measured values to other collections from the region and to values predicted by the OIPC supports the assessment that our samples are a reasonable indication of the isotopic composition of snow in this study area. We make this point in our revised text (Lines 256-259, 335-336, 400-401)*

L389-395. the fact that this potential bias in the data (originated by the road) is brought up this late in the paper seems problematic to me. Are these data included in all the previous analysis? Why? should this data be discarded altogether?
*-We agree that the possibility that road dust influenced the precipitation samples collected at the spring site is concerning. This is why we relied on the precipitation collected at RG-2 (far from any road) to constrain precipitation chemistry (Lines 411-413).*

L403. The authors should discuss why this significant estimated contribution from RG is not reflected in the stream isotopic composition time series, which remains stable although there's a strong trend in the RG series (figure 6).
*-In reality, only the spring (groundwater) time series is stable (Figure 6); the streamwater exhibits rising isotope values late in the season that are consistent with an increased rock glacier component.*

L413. but you also estimated 15 l/min for one of the glaciers. So, the rate of ice melt is hugely uncertain! Although this surface depression analysis is interesting, the way it is presented here feels rushed and somewhat contrived. The authors should expand: what is the total area of RG mapped in this basin? What is the range of glacier ice available? etc.
*-We previously discussed our logic in presenting the implications of this surface depression so late in the discussion. We added mention of the total area of rock glaciers mapped in the basin (Line 81). We previously mentioned that glaciers are absent in this region. (Line 77)*

Figure 12. I'm afraid that I can't see the depression the authors refer to. This data should be presented much earlier, in the data section, and not in the discussion section.
*- As we have discussed already, given the uncertainties we feel it is prudent and appropriate to include description of this depression and its possible significance late in the discussion section.*

*To aid clarity and to highlight the depression we refer to in the text, we added contour lines to the figure and changed the color of the oval call-out to yellow.*

L428. Based on only one sample of snowmelt water, it is tenuous to make strong statements about snowmelt similarity or influence on streamflow throughout the season. This is a major problem of the material presented here, and should be discussed by the authors.

*-As noted above, we added a more explicit statement about the limitations imposed by the small number of snow samples (Line XXX). In this part of the discussion, where we explore the rock glacier contribution to streamflow late in the melt season, the composition of the snow is not an issue because the calculations are based off the triangle defined by the other end members: rain, groundwater and rock glacier water (Figure 11).*